# Epidermal Wearable Biosensors for the Continuous Monitoring of Biomarkers of Chronic Disease in Interstitial Fluid

**DOI:** 10.3390/mi14071452

**Published:** 2023-07-20

**Authors:** Xichen Yuan, Oumaima Ouaskioud, Xu Yin, Chen Li, Pengyi Ma, Yang Yang, Peng-Fei Yang, Li Xie, Li Ren

**Affiliations:** 1Key Laboratory of Flexible Electronics of Zhejiang Province, Ningbo Institute of Northwestern Polytechnical University, Ningbo 315103, China; xichen.yuan@nwpu.edu.cn (X.Y.); oumaimaouaskioud1999@yahoo.com (O.O.); licc@mail.nwpu.edu.cn (C.L.); 2019303577@mail.nwpu.edu.cn (P.M.); yangpf@nwpu.edu.cn (P.-F.Y.); 2MOE Key Laboratory of Micro and Nano Systems for Aerospace, School of Mechanical Engineering, Northwestern Polytechnical University, Xi’an 710072, China; yin_xu@mail.nwpu.edu.cn; 3Key Laboratory for Space Bioscience and Biotechnology, School of Life Sciences, Northwestern Polytechnical University, Xi’an 710072, China; xieli@nwpu.edu.cn; 4Ministry of Education Key Laboratory of Low-Grade Energy Utilization Technologies and Systems, Chongqing University, Chongqing 400030, China; yang_yang@nwpu.edu.cn

**Keywords:** interstitial fluid, epidermis, wearable biosensor, chronic disease

## Abstract

Healthcare technology has allowed individuals to monitor and track various physiological and biological parameters. With the growing trend of the use of the internet of things and big data, wearable biosensors have shown great potential in gaining access to the human body, and providing additional functionality to analyze physiological and biochemical information, which has led to a better personalized and more efficient healthcare. In this review, we summarize the biomarkers in interstitial fluid, introduce and explain the extraction methods for interstitial fluid, and discuss the application of epidermal wearable biosensors for the continuous monitoring of markers in clinical biology. In addition, the current needs, development prospects and challenges are briefly discussed.

## 1. Introduction

Nowadays, with the growth of the world’s population and the prolongation of life spans, chronic diseases have become a significant and long-lasting trend. According to the report from the World Health Organization (WHO) in 2018, four major chronic diseases (cancer, cardiovascular diseases, chronic respiratory diseases and diabetes) account for 71% of all human deaths globally every year. The convenient, rapid and accurate monitoring of critical disease biomarkers will become crucial for chronic diseases diagnoses, treatment and health care.

Wearable biosensors are smart, miniaturized devices that have proven high performance in the real-time analysis of various parameters and in extracting and monitoring physiological and biochemical signal features. They can provide crucial information regarding the real-time situation of the wearer, and will become an attractive and competitive choice due to their significant properties of low cost, portability and sensitivity for monitoring chronic diseases. Unlike the clinical standard for patient health tracking, which is based on extracting blood through invasive and uncomfortable techniques at discrete times, wearable biosensors can detect and monitor biomarkers and therapeutic drugs from bodily fluids (interstitial fluid (ISF), sweat, saliva and tears) in a noninvasive/minimally invasive and continuous manner [1,2,3,4].

Wearable biosensors have been extensively explored for the monitoring of ISF, sweat, saliva and tear biomarkers. Saliva is an attractive body fluid because it is composed of a rich matrix of constituents (e.g., drugs, hormones, metabolites or antibodies), secreted in a continuous manner, and because it can be easily sampled [5,6]. However, wearable biosensors in the buccal cavity (e.g., a biosensor integrated in a mouthguard) are confronted with many challenges; a sensor with a highly sensitive, specific and stable performance is needed due to the extremely low concentrations of biomarkers and high concentrations of interferential protein (mucins and proteolytic enzymes), contamination (by external factors from food, drink, etc.) and a high-moisture environment [1,7,8,9,10] as well as the requirement for wearable saliva biosensors to undergo further human clinical trials to confirm reliability [1,6]. Tears contains more than 20 different species (e.g., electrolytes, proteins, metabolites, trace metals) and various metabolites exhibit a close correlation between their concentrations in tears and blood (e.g., glucose) [1,5,10]. Contact lenses used for vision correction and cosmetic reasons touch the cornea in a noninvasive manner, holding great potential to serve as an ideal sensor platform for the real-time and continuous monitoring of tear biomarkers [9,11]. However, tear-based biosensors primarily focus on glucose monitoring, and none of them have been successfully reported in long-term clinical trials in the human eye [11]. Epidermal wearable biosensors represent an exciting area of research in the field of wearable technology for chronic disease monitoring [12] and have been widely used in various parts of the body (e.g., the arm, forehead, chest and back) for sweat and ISF biomarkers monitoring [3,12]. They are manufactured on a variety of substrates (textiles [13], wristbands [14], smart bandages [15], temporary tattoos [16], etc.) and are convenient and comfortable to wear [3,12]. Although sweat contains important biomarkers (e.g., ions, proteins, metabolites, drugs), the analyte concentration in sweat is highly different from that in the blood, and the protein content in sweat is usually more than 1000 times lower than that in the ISF and the blood [12]. A deep understanding of the correlations between sweat analytes and health status is extremely important for the development of sweat biosensors [1,12].

ISF is a biological extracellular fluid, formed by transcapillary exchange/infiltration during blood flow and the transport of nutrients and wastes among cells, blood and lymphatic capillaries [3,17,18]. It is present in most of the dermis, and is also around the salivary glands and sweat glands [8]. ISF contains important biomarkers, such as ions (e.g., Na^+^, K^+^, Ca^+^ and Cl^−^), small molecules (e.g., glucose, lactate, uric acid, peptides and ammonia) and proteins. Compared to other biofluids (sweat, saliva and tear), ISF has several advantages for wearable biosensing [3,19,20,21,22]. Firstly, the composition and temporal profiles of ISF are close to blood analysis; secondly, there is a much lesser dilution of biomarkers in the ISF; and thirdly, ISF is free of blood cells and other clotting factors. All of these characteristics make ISF a potentially useful biological matrix for long-term use and the simple and continuous monitoring of biomarkers with more stability and reliability in sensor dynamics, which is extremely important for chronic disease diagnosis and management.

In this review, we introduced the composition and characteristics of ISF and the preclinically explored ISF biomarkers of chronic disease. Then, we explained the ISF extraction and analysis methods. Moreover, the application of epidermal wearable biosensors in the continuous monitoring of preclinical biomarkers was investigated, and related developments were introduced and discussed. Finally, the challenges and development potential of epidermal wearable biosensors for the continuous monitoring of clinical biomarkers were summarized.

## 2. ISF Characteristics

Sweat, tears and saliva have been extensively studied as potential sources of bodily fluids for biosensing (Table 1). Moreover, a significant focus on developing wearable subcutaneous devices has been increasing in the last several decades to create devices that can monitor ISF located beneath the epidermal layer. Although the composition and source of ISF in the skin are difficult to determine, its diagnostic potential comes from its relatively easy access, and the analytes in it correlate well with gold-standard blood sampling [3,4,23,24].

Consequently, as ISF contains important ions such as Na^+^, K^+^ and Cl^−^ as well as metabolites like glucose and lactate, and plays major roles in organ regulation and homeostasis, miniaturized wearable devices that can be used for the real-time sensing of ISF are already available and commercialized, such as Abbott’s FreeStyle Libre and Medtronics’ iPro Evaluation system, which are used to monitor glucose levels in diabetic patients [75].

In living skin tissues, skin cells are surrounded by ISF [17], and the small and uncharged molecules (e.g., cortisol) directly diffuse from the capillary endothelium into ISF and maintain the diffusion balance between ISF and blood vessels [24]. On the other hand, large and charged analytes (e.g., proteins, glucose) mainly traverse directly through the space between cells, or transport through vesicles (as shown in Figure 1) [3,8]. There is a much lesser dilution of biomarkers in the ISF. This feature leads to the correlation between the concentration of many biomarkers in blood and ISF, such as electrolytes (Na^+^, Mg^2+^, Ca^2+^, K^+^ and phosphate, etc.), metabolites (glucose, lactic acid, cortisol, etc.), protein, etc. (as shown in Figure 1) [23,24,65,76]. Studies have found that the average concentration of Na^+^ in blood is 141.2 mM, and the average concentration of Na^+^ in ISF is 135.7 mM; the average concentration of K^+^ in blood is 4.37 mM, and the average concentration of K^+^ in ISF is 3.97 mM [23]. Similarly, for glucose, lactate, etc., there are also a lot of data showing that the concentrations in blood and ISF are almost the same. However, for cortisol, penicillin, morphine and other drugs with short half-lives, the correlation is weak. Studies have shown that the total blood concentration of cortisol is between 80 and 500 nM with day and night changes, but its concentration in ISF is 5–50 nM [68]. After the injection of penicillin G into sheep, the blood/ISF concentration ratio was 53 mg/kg, and after 2 h, it had not reduced to similar levels [77,78]. For higher-molecular-weight analytes (such as proteins and lipids), the ISF/blood concentration ratio has an antilog relationship with the molecular weight [79], so conversion factors are required in the application.

Research on wearable health monitoring initially concentrated on the needs of physical sensing. This led to the creation of portable physical sensors, which are electronic devices that integrate sensors into or with the human body to access, monitor, calculate and analyze biophysical signals such as heart rate, skin temperature, respiration rate and brain activity. These devices can take many forms, including tattoos, gloves, clothing and implants [75,83]. Recently, advances in printed electronics and materials have allowed flexible sensors to be even smaller and worn as skin patches [84]. These biosensors are also able to detect changes in pH levels, glucose and ions in the human body [19,85,86,87]. In addition, wearable biosensors can offer direct information on specific disease biomarkers and metabolite changes in bodily fluids to provide a continuous, real-time monitoring of various physiological parameters to improve the accuracy of diagnosis and disease recording [5,58,61,88].

At present, the applications of wearable biosensors in ISF are mostly focused on: glucose [6,10,86], cortisol [65,89], urea [5,90,91] and lactic acid [19,20,87,92,93]. Of course, the study of other biomarkers is also of great significance. It can be extended to the areas of the evaluation of a series of protein disease markers [21], hormones [61] and stress markers [94], and it may also provide new insights into circadian rhythms and disease trajectories by evaluating the dynamic concentration fluctuations of the biomarkers in different scenarios [22]. As research on ISF wearable biosensors delves deeper, researchers have also discovered some drawbacks. When the lymphatic system continues to clear the ISF, the liquid pressure is negative relative to the atmospheric pressure, about 500–1000 N·m^−1^, and this negative pressure would complicate sample extraction [95]. It is also uncertain whether ISF samples can be reliably extracted through needles or skin perforations without changing the analyte concentration. In addition, when using wearable biosensors to analyze epidermal ISF, since most of the dermis is acellular and its metabolism is slow, there is an ISF hysteresis phenomenon in the exchange process between blood and ISF in the dermis [8]. Therefore, during continuous monitoring, a corresponding design is needed to minimize the equipment delay to increase the credibility of the results.

## 3. ISF Extraction Methods

ISF filtration has been documented since the 1980s when Starling discovered the exchange of metabolites and electrolytes between blood plasma and the interstitial compartment through the endothelial cell wall. In recent years, ISF analysis has been applied to detect metabolites and a variety of biomarker diseases, such as cancer [58,96] and chronic kidney disease (CKD) [5,97]. Wearable biosensors have sparked a great interest for accessing ISF in a non-invasive, non-contaminated and efficient manner [98,99,100].

The Wick method was one of the first methods used for ISF extraction; the concept is to insert an absorbent wicking nylon material of 0.1 mm into the skin, saturate it with ISF, and pull it out for analysis [24,101]. Although this technique shows an adequate equilibration with ISF [24], it is slow, invasive and provokes an inflammatory reaction in the insertion site [101].

Another technique that is applied to harvest the ISF is the suction blister fluid method, which realizes a high extraction efficiency. First, micropores are generated within pretreated skin, either via ultrasound, laser or other techniques, and then the ISF is collected in a vacuum by applying a negative pressure to the skin [24,79,80]. Thus, a high degree of tissue damage is incurred, affecting the concentrations of the analytes, especially large-molecule analytes.

The microdialysis method requires inserting a small microdialysis catheter into the skin [102,103,104,105,106]. The catheter has a semi-permeable membrane that allows small analytes (e.g., glucose, ions) and proteins (e.g., albumin) to exchange with the liquid in the probe, which can then be extracted and sensed [107,108,109,110,111,112]. This method is based on a passive diffusion process. More recently, many studies have focused on improvements in the wearability of this platform, but the MD method takes a long time to sample, increases the hysteresis of the ISF and the equipment is large in size and can cause long-term skin irritation. These shortcomings make the method less viable for wearable biosensing than other minimally invasive indwelling sensors [24,75,113].

In order to carry out real-time continuous monitoring, advanced methods are required for ISF collection and sampling. Reverse iontophoresis (RI) can be used for the non-invasive extraction of ISF from the body and for performing in situ real-time detection (as shown in Figure 2) [20,114,115]. There is another very attractive method, microneedle arrays (MNAs), which mini-invasively destroys the skin and forms a short fluid path (about 500 μm) (as shown in Figure 3); the analyte diffuses from the ISF to the adjacent stereo sensor for detection [3,4,98,116,117,118,119]. In addition, an indwelling method that resides in the tissue can also be used by being mini-invasively implanted and immersed in the ISF for detection (as shown in Figure 4). Of course, the method of extracting ISF is not limited to these techniques; some other unique methods are also under development [120].

### 3.1. Reverse Iontophoresis

Normally, the skin surface is negatively charged but tissue fluid is mainly constructed of Na^+^ and Cl^−^ ions. The concept of the RI method for ISF extraction uses electric potential between the anode and cathode; sodium ions migrate toward the cathode and generate the electric current, as a consequence, and chloride ions migrate toward the anode [4,24,121]. The voltage applied across the skin induces an electroosmotic flow of ISF from the anode to cathode, forming a moving sheath of sodium ions through the paracellular route [24] and leading to the electroosmosis of the neutral molecules’ (e.g., glucose) transmission (as shown in Figure 2A,B) [37,115]. Then, the neutral molecules can be collected at the cathode and directly measured with a traditional sensor placed at the cathode (as shown in Figure 2B) [37]. Anionic molecules (e.g., lactate) will flow toward the anode, and can then be collected and quantified with a biosensor located at the anode (as shown in Figure 2C) [20]. The RI method leads to the molecules’ movement out of the skin with no harm, invasiveness or blood contact [1,122,123].

The main driving force for RI-based biosensors is related to glucose monitoring (as shown in Table 2) [37,115,124,125,126]. GlucoWatch biographer (Cygnus, Inc., Redwood City, CA, USA) was the first commercial non-invasive glucose sensor platform approved by the U.S. Food and Drug Administration in 2001 [127,128,129]. It has shown great potential and a high ability to control and measure glucose concentration continuously and frequently [128]. Garg et al. [129] compared and confirmed the correlation between GlucoWatch biographer glucose values and capillary blood glucose values obtained by the HemoCue analyzer in the clinical setting and the One Touch Profile meter in the home setting. Unfortunately, the device was withdrawn in 2007 due to reported skin irritations and reproducibility issues [123,130]. However, the RI-based ISF extraction concept stayed on and the research on improved RI techniques for glucose sensing has expanded. Kim et al. developed a cellulose/β-cyclodextrin (β-CD) electrospun immobilized glucose oxidase enzyme patch for the noninvasive monitoring of ISF glucose levels, and high-accuracy RI was carried out by applying a mild current with two skin-worn electrodes to noninvasively uptake glucose from ISF [124]. Yao et al. demonstrated a two-electrode non-invasive ISF glucose sensor for stability and the continuous monitoring of glucose levels, and the extraction of ISF through the RI process and the detection of glucose concentration through an amperometric approach was conducted with the same two electrodes [125]. Xu et al. developed a conductive hydrogel-based electrochemical biosensor incorporated with RI via the in vivo noninvasive and continuous monitoring of ISF glucose, which showed a good correlation with the finger-stick blood test using a glucometer [126].

In addition, RI has been used for monitoring a multitude of biomarkers other than glucose (e.g., lactate) (as shown in Table 2). Lactate is a byproduct of anerobic glycolysis and is an important biomarker for determining oxidative stress levels, muscle health and tissue hypoxia [20]. Due to its anionic nature at physiological pH, extracting lactate from ISF via RI relied on a high current density and a longer RI time (~hours) [131,132]. Such an extended current application might lead to damage to the skin surface. De la Paz et al. developed a flexible, skin-worn device that integrates an RI system and an amperometric lactate biosensor placed on the anodic electrode for simultaneous ISF lactate extraction and quantification, respectively. Using this integrated device, rapid lactate collection from the ISF can be realized after 10 min of RI with no evidence of discomfort or irritation to the skin [20].

The RI has also been a useful technique for the non-invasive monitoring of amino acids [133], cortisol [89] and other biomolecules in the body that are collected. It can also be used to extract small analytes (urea [90], phenylalanine [134], valproate [135], etc.) for the continuous monitoring of human health, as well as for drug monitoring. Moreover, the movement of molecules from the dermis to the epidermis is affected by many factors, such as the diffusion of molecules, the fat and moisture content of the skin and the physiological pH [20,130]. Since the diameter of the follicular channel is the largest, it has the least resistance and is the preferred channel for molecules [130], and so, the liquid collected by this method is filtered ISF, not pure ISF. Lipani et al. [114] developed a graphene glucose-monitoring platform based on path selection (hair follicle, intercellular or transcellular pathway); the platform is composed of graphene sensors and hydrogel reservoirs and the ISF is drawn into it by electroosmosis, which improves the consistency of analyte extraction by RI.

### 3.2. Microneedle Arrays

The development of new transdermal ISF extraction methods has attracted widespread attention. MNAs are the miniaturization of conventional hypodermic needles, and their height is about several hundred micrometers [3,98,136]. They are inserted into the dermis, applying pressure adjacent to the microneedles (MNs) (as shown in Figure 3) [24,137]. They were originally used for transdermal drug delivery [138], and they were first developed by Prausnitz et al. to study the transdermal permeation of drugs and vaccines and the movement of molecules across the stratum corneum [139]. Because of their extremely small size, they can avoid stimulating dermal nerves or destroying dermal blood vessels to reach the ISF [116,137,140,141].

In the past decade, many studies have focused on achieving the continuous clinical monitoring of disease-related biomarkers (such as glucose, lactic acid, glutamate, etc.) [142,143,144], and also therapeutic drug monitoring to measure the concentration of administered medications and analyze their metabolic characteristics in order to guide drugs’ administration and doses and to apply pharmacokinetic principles for efficient therapy [22,88,145,146]. There are many types of MNAs that have been developed, such as dissolvable, coated, hollow, solid and porous MNs [17,61,86,140,147,148,149].

Various materials are now used to prepare MNs with different shapes, sizes, morphological characteristics and densities [141,150,151,152], which are used to manufacture MNAs and applied to the extraction and analysis of human ISF. The MNAs devices, which are made for ISF analysis are mainly based on hollow and solid MNs, and they are fabricated from either silicon, metal, hydrogel or polymer (as shown in Figure 3 and Table 2).

Ultimately, every kind of MN absorbs ISF in different ways, with hydrogel MN swelling to allow diffusion [21] and hollow MNs aspirating ISF under capillary action [86]. The choice of MN material is crucial for the ISF extraction volume and rate, with GelMA MN, sponge-forming poly(vinyl acetal) MN and cross-linked GelMA MNs being among the materials used [140]. Researchers have also used external forces to extract ISF faster, which can be by preparing an MN patch combined with the reverse iontophoresis method to provide electroosmotic force [141]. Various techniques have been implemented for enhancing ISF collection following MN penetration, such as attaching filter paper or ultrafine MNs to the MN’s base. Despite progress in increasing the ISF extraction volume and decreasing the extraction time, transferring the extracted ISF to another analytical instrument still takes time, and sample stability may be insufficient [145].

In recent studies, Singamaneni et al. demonstrated that MNAs can be used to selectively capture biomarkers on the skin surface, and an on-needle immunoassay can measure the captured biomarkers with high sensitivity using plasmonic-fluor, which is a bright fluorescent nanostructure that enhances the detection limit of protein biomarkers [153,154]. Overall, MNAs optimized selectivity and sensitivity for the detection and quantification of protein biomarkers. MNAs have the characteristics to solve most of the problems faced by current methods and are the most promising method, combined with processing technologies for ISF extraction and continuous monitoring sensors, with integrated monitoring and delivery functions used for fluid collection, and physical parameters and biomarkers’ diagnostic and cosmetic therapies and medication analysis [116,145]. MNAs can also be attached to a 3D printed device that has a tilted angle to allow the penetration of MNs at an oblique angle, which increases the surface area of contact and improves ISF extraction [155]. This method was confirmed to extract a larger volume of ISF compared to traditional microneedle devices and to be more consistent for the reproducible extraction of ISF.

For the extraction of ISF using the MNAs method, the main challenges in designing MNAs are the biocompatibility and avoiding the fragility of MNs. It is understood that, in order to overcome these problems, various types of materials and methods have been tested to provide suitable mechanical strength and maintain the good biocompatibility of MNAs, for example, a combination method which uses solid-state MNAs to coat nanomaterials [156].
Figure 3Microneedle arrays for ISF extraction. (**A**) Schematic illustration of stainless steel MNs (ssMNs) surfaces precoated with ZnO nanowires (NWs) and poly(vinyl pyrrolidone) (PVP) for the protecting and electrochemical sensing of subcutaneous H_2_O_2_ (reprinted with permission from [147], copyright 2019, American Chemical Society). (**B**) Schematic illustration of hollow MN-based continuous glucose-monitoring (CGM) device for transdermal ISF glucose detection, inserted SEM image is the hollow MN (reprinted with permission from [144], copyright 2023, American Chemical Society). (**C**) Stainless steel MNA-based touch-actuated glucose sensor: skin penetration using solid MNAs to create microchannels in the skin, then glucose extraction using RI from pierced skin; inserted SEM image is an MN (reprinted with permission from [137], copyright 2022, Elsevier). (**D**) SEM image of Si–MNA and optical image of the Au–Si–MNA electrode used for the electrochemical immunosensor of breast cancer biomarker detection (reprinted with permission from [58], copyright 2021, Elsevier). (**E**) SEM image of polyester MNA and optical image of the stretchable MNA-based biosensing platform for the real-time wireless monitoring of sodium levels in ISF (reprinted with permission from [157], copyright 2022, John Wiley and Sons). (**F**) Hyaluronic acid hydrogel-based MNA tattoo for the simultaneous colorimetric detection of four biomarkers (i.e., pH, uric acid, glucose and temperature) in vivo and the SEM image of the MNA (reprinted with permission from [85], copyright 2021, John Wiley and Sons). (**G**) Schematic images of the swellable methacrylated hyaluronic acid hydrogel MN for levodopa sensing-based Parkinson management, and its swelling behavior in the in the gelatin phantom in 3 min (reprinted with permission from [22], copyright 2023, Elsevier).
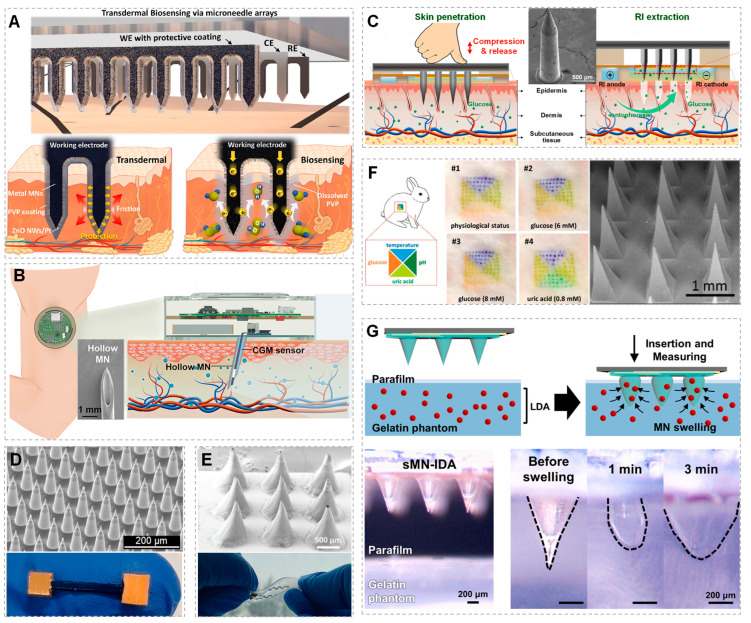


### 3.3. Indwelling ISF Sensors

Indwelling ISF sensors are much simpler than those that need to extract ISF (as shown in Figure 4A), and some indwelling sensors have been approved by the market [158]. For example, a FreeStyle Libre Flash glucose-monitoring system was created by Abbott, which has a three-electrode system integrated into a large needle that was inserted using a spring-loaded inserter. This small sensor can be worn on the back of the upper arm (as shown in Figure 4B), soaked in ISF about 5 mm below the skin surface and continuously detect glucose in ISF for 14 days, and the glucose reading can be obtained using an electronic device with near-field communication. The device provides current glucose levels and an 8 h trend graph without needing external calibration with finger-prick blood glucose detection [159]. The results show that there is consistency between the sensor readings of the system and the venous reference values, which proves the accuracy of the FreeStyle Libre Flash glucose-monitoring system [160], laying the foundation for the indwelling ISF sensing system to be used in the field of health monitoring. The FreeStyle Libre system was approved in Europe in 2014, and in the United States (US) for professional use in 2016 and for personal use in 2017 [161]. It is a powerful and successful technique for the continuous and long-term monitoring of substances in a variety of tissues. It allows long-term continuous sampling as well as the manipulation of local metabolism with minimal tissue damage [162].

The FreeStyle Libre system and other commercial systems (as discussed In “Section 4. ISF Sensing Platform for Continuous Monitoring”) are based on a needle-type sensor for subcutaneous ISF extraction and glucose detection [151]. Other researchers have also developed implantable sensors for long-range continuous glucose monitoring (as shown in Figure 4A) [163], for example, Hassan et al. developed a fully passive miniaturized circuit composed of an inductor–capacitor tank resonator with a volume of 16 mm^3^; this circuit can be implanted under the human skin, where the ISF surrounds the inductor–capacitor tank resonator, and the variations in glucose concentration can be monitored [164]. Jin et al. presented a continuous glucose-monitoring platform consisting of a signal conditioning part, a programmable electrochemical chip and a wireless connection using Bluetooth low energy with a smartphone (as shown in Figure 4C) [165]. There is a reliable correlation between the ISF level and the blood glucose level; and thus, implantable biosensors for the continuous monitoring of ISF glucose (as shown in Table 2) are considered to be the next-generation products to replace traditional glucose meters [151,163,165,166].
Figure 4Typical indwelling ISF sensors. (**A**) A common configuration of the transdermal glucose biosensor applied to the skin (reprinted with permission from [163], copyright 2023, Elsevier). (**B**) A camera image of the Abbott Freestyle Libre 3 Flash glucose-monitoring system [167]. (**C**) Diagram of the implantation of glucose sensor and the biosensing mechanism (reprinted with permission from [165], copyright 2022, Elsevier).
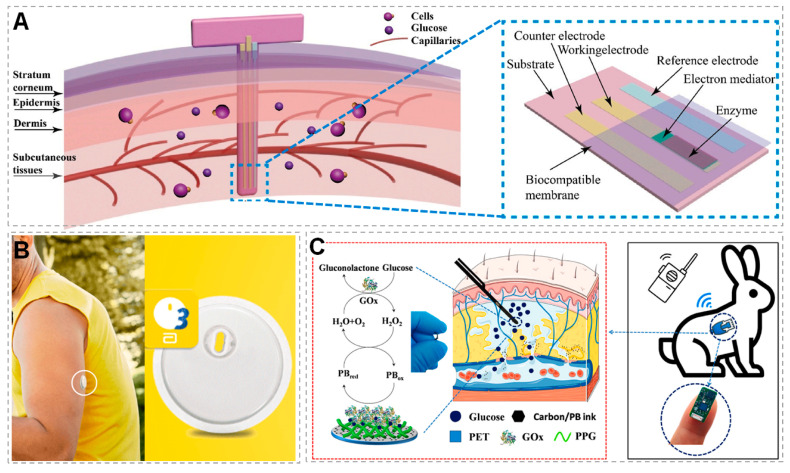

micromachines-14-01452-t002_Table 2Table 2Summary of ISF extraction methods.ISF Extraction MethodsParameters/Related materialsLasted TimeDetected BiomarkersReferenceRICurrent density = 0.4 mA/cm^2^10 minLactate[126]Current density = 0.3 mA/cm^2^5 minGlucose[115]Potential = −3 V10 minGlucose[126]Current density = 2 mA/cm^2^Potential = 2~5 V5 minGlucose[125]Current density = 0.27 mA/cm^2^3 minGlucose[37]Current density = 0.5 mA/cm^2^5 minGlucose[114]Current density = 0.3 mA/cm^2^15 minGlucose, Lactate[131]MNAsHydrogel MNA (MeHA ^1^)Conic shapeHeight: 600 μmBase diameter: 400 μmCenter-to-center distance: 600 μm3 minLevodopa, Dopamine[22]Hydrogel MNA (MeHA ^1^)Height: 850 μmThe base of each needle away from its neighbor: less than 250 μm5 minGlucose, uric acid, insulin, serotonin[21]Metal MN24-gauge hollow MNHeight: 2 mm2.7 minGlucose[144]Metal MNAHeight: 1200 μmBase diameter: 480 μmCenter-to-center distance: 1500 μm10 minGlucose, Na^+^, K^+^[146]Polymer MNA coated with goldSquare-based pyramidHeight: 1000 μmBase width: 500 × 500 μmThe base of each needle away from its neighbor: 500 μm30–40 minLactate[143]Silicon MNAHeight: 250 μmBase diameter: 50 μmCenter-to-center distance: 110 μm60 minErbB2 ^2^[58]Indwelling ISF sensorsImplanted electromagnetic sensorRectangular outer ring: 11.6 × 14.6 mmSquare inner ring: 11 × 11 mmPolyamide substrate: 12 mm × 15 mmContinuousGlucose [168]Implanted circuitInductor–capacitor tank resonator: 16 mm^3^ContinuousGlucose[164]Implanted electrochemical biosensorFlexible electrode componentsEnzyme sensing layerPolyurethane outer layerMiniaturized printed circuit boardContinuous for 30 daysGlucose[165]^1^ MeHA: Methacrylated hyaluronic acid. ^2^ ErbB2: Epidermal growth factor receptor 2.

## 4. ISF Sensing Platform for Continuous Monitoring

For many years, ISF has been used for the non-invasive diagnosis of metabolic disorders and the evaluation of treatment effects and organ failure [169]. As far as the wearable epidermal biosensor being developed is concerned, people’s efforts have mainly been focused on the continuous monitoring of biomarkers such as glucose and cortisol [170], because these biomarkers are the most common in clinical applications and the study of their correlation with blood is relatively mature.

Wearable biosensors were first used for the monitoring of physical parameters such as temperature, calories and heartbeat [171]. The industry and market witnessed a huge rise in this wearable technology, and the most representative example is the GlucoWatch sensor [172]. This device uses RI to extract ISF for continuous blood glucose monitoring. It has been approved for commercial applications, but due to reported skin irritations and reproducibility issues [123,130], the device was finally withdrawn. Afterward, new products continued to emerge, including products such as Google glasses, Apple watches, Xiaomi bands, wristwatches, chest patches, and other smart clothing items that were developed by various companies for monitoring the wearer’s health. Recently, these products have been developed to become smarter, more miniaturized and able to be worn as skin patches to monitor lactate, glucose, tyrosinase, cancer-related enzymes and antibodies like anti-SARS-CoV-2 IgM/IgG antibodies. Furthermore, their monitoring abilities include blood pressure (iHealth), activity and sleep (iHealth, Fitbit, Apple, and Garmin), pulse oximeter (iHealth and Nonin Medical), cardiovascular health (Hexoskin, Zephyr strap, MC10 BioStamp) and glucose (GlucoWatch G2 Biographer, GlucoTrack, Abbott Freestyle Libre 2, Johnson and Johnson, Roche, and Dexcom G5 and G6) [151,171,173,174]. Early CGM sensors required frequent calibration, but newer models like Dexcom G6^®^ and Abbott FreeStyle Libre no longer require calibration, enabling “zero-finger pricking” glucose monitoring. While initially classified as aids in detecting hypo- and hyperglycemic episodes, Dexcom G5^®^ received FDA approval in 2016 to replace the self-monitoring of blood glucose (SMBG), and Dexcom G6 and Abbott FreeStyle Libre can now also serve this purpose. In 2018, the FDA introduced a new classification called an integrated CGM system, placing it in the moderate-risk class II category. This regulatory change reduces the burden for iCGM devices, allowing them to transmit glucose-monitoring data to digitally connected devices for managing diabetes, raising their commercialization applications [175].

Subcutaneous implantable CGM devices are also interestingly useful and have been commercialized. There are seven FDA-approved and commercially available implantable glucose sensors with six of them utilizing electrochemical enzymatic sensing [176]. And one new system called Eversense, based on non-enzymatic fluorescent methods, was also approved by the U.S. Food and Drug Administration (FDA) on 6 June 2019, and has been available in the European Union and European Economic Area since May 2016 [4]. These devices are easier to use and use advanced communication functions to connect sensors to portable smart devices for glucose concentration tracking. However, this type of sensor has problems, such as a short life and difficult sensor calibration. Generally, these subcutaneous devices need to be replaced every 3–7 days and recalibrated every 12 h [177].

Transdermal biosensors, especially the ones which are based on the RI extraction method, are more convenient because of their flexibility and pain-free characteristics [115]. Cheng et al. [137] combined the RI system with MNA and electrochemical glucose detection to prepare a wearable biosensor, which has a higher selectivity and a more accurate analysis results. Chen et al. [178] used a flexible biocompatible paper battery to design a wearable ISF biosensor with a combination of ETC (electrochemical twin channels) and RI. Human clinical trials were also carried out. Continuous measurements were carried out on human subjects within one day. The results showed a good correlation between ISF and blood glucose levels, which opened up a new perspective for clinically non-invasive continuous blood glucose monitoring.

MD has also been proven to continuously measure the glucose concentration in the ISF in a self-monitoring glucose sensor, and it has a good correlation with blood glucose levels [179]. However, the MD method has a long lag time, and the probe is prone to scaling or degradation when used for continuous monitoring. The MN method avoids the problem of the molecular weight limitation of the MD method, and combines it with microprocessing technology, making it easier to integrate with the sensing component, so it is more suitable for continuous monitoring [180]. The typical application of the MN method in ISF sensing is in the glucose sensor designed by Zimmermann et al. [181] to extract ISF by capillary force, but it fails to continuously extract ISF for continuous monitoring. Recently, Coffey et al. [182] used biometric probes to modify the surface of the MNs, which increased the selectivity for the target protein and proved to be applicable to accumulate and detect low-concentration analytes over a longer period of time [183]. Pu et al. [184] proposed a wearable flexible electrochemical sensor with three electrodes on a PDMS microfluidic chip, which could be used for the extraction, collection and detection of ISF, so as to achieve continuous glucose monitoring. The device uses inkjet printing to modify graphene on the surface of the working electrode and gold nanoparticles on the graphene layer to achieve a high-sensitivity and low-concentration glucose detection, with a detection limit of 0.3 mg/dL, which has the potential for the clinical detection of hypoglycemia.

More recently, tattoo ink has also been used for optical biosensing in ISF. The tattoo ink contains biosensors to detect specific analytes such as glucose and lactate. It is then injected into the skin using standard tattooing techniques, to detect specific biomolecules. The technique has shown many advantages over traditional biosensors, including long-term stability, improved sensitivity, and the ability to detect multiple analytes simultaneously. In addition, ink biosensors can also be read using standard optical imaging techniques, which makes them accessible and easy to use [185].

It can be seen that the continuous monitoring of biomarkers based on ISF has great development prospects, but continuous skin irritation is a problem that still needs to be improved. In addition to that, the hysteresis effect in the sensing process will increase the inaccuracy of the results. Therefore, it is necessary to solve the current limitations in order to better develop the potential of ISF wearable biosensors.

## 5. Clinical and Preclinical Applications

GlucoWatch [172], a glucose-sensing device, is the most typical application of epidermal wearable biosensors in ISF. Although it was eventually eliminated by the market, it laid a deep foundation for ISF sensors. Yuen et al. [186] functionally modified the silver film on the surface of the nanosphere with a self-assembled monolayer film and implanted it under the skin of rats to monitor the glucose concentration in the ISF continuously and in real-time. This led us to believe in the potential of wearable biosensors using ISF as samples. Freckmann [187] and Mian et al. [170] outlined the current available continuous glucose-monitoring equipment, which could monitor the glucose level in the ISF of patients with glucose disease for 6–14 consecutive days in clinical settings. Bruttomesso et al. [188] also investigated the clinical application of real-time ISF glucose monitoring and intermittent ISF glucose monitoring, and the results showed that people strongly believe that monitoring methods can reduce the risk of hypoglycemia and improve treatment satisfaction, and that they are superior to the self-monitoring of blood glucose through finger pricks.

In recent years, the health detection of ISF sensors in other biomarkers has been gradually emerging. Arya et al. [189] used an electrochemical impedance method to detect cortisol in ISF using bifidobacteria modified with interdigital microelectrodes; Parrilla et al. [149] modified the solid MNs with different coatings, and designed an epidermal patch for the continuous monitoring of the K^+^ concentration changes in ISF. Important progress has been made in research into the intradermal analysis of electrolyte balance-related examples, which can be applied to clinical disease analysis. Similarly, Bollella et al. [92] reported an MN-based biosensor for the minimally invasive and continuous monitoring of lactic acid in ISF, and developed a wearable biosensor based on a painless MNA on this basis to simultaneously continuously monitor lactic acid and glucose in ISF [44]. These results all demonstrate the potential of new biosensors based on MNAs in sports medicine and clinical nursing applications. Moreover, Parrilla et al. [86] used a hollow microneedle sensing patch that can be attached to the skin. The microneedle patch has an electrode inside that can detect glucose levels in the interstitial fluid directly below the skin surface. The device is able to detect low glucose concentration (Table 3), and it is designed to be worn for long periods, making it suitable for continuous glucose monitoring. Recent advances in micro- and nanofabrication technologies have taken part in the development of more biosensors with transdermal sensing platforms for the real-time monitoring of ISF analytes, including pH levels in the ISF of tissues and organs. Dervisevic et al. [18] described a microneedle array-based sensor that can be attached to the skin to monitor pH levels in ISF. The sensor is made up of a polymer microneedle array that is coated with a pH-sensitive hydrogel. The hydrogel changes color in response to changes in pH, and the color change is detected using a smartphone camera. Monitoring multiple biomarkers in ISF has become a novel feature for wearable biosensors; a microneedle array was first integrated with sensing elements to measure multiple biomarkers. The microneedles are made of a biocompatible polymer and are coated with sensing elements that can detect specific biomarkers in the ISF, such as glucose, lactate and pH level [190].

## 6. Summary and Prospect

Over the last decade, epidermal wearable biosensors have shown great potential to revolutionize the solutions and methods of managing healthcare by providing a wealth of data and insights for the continuous, real-time monitoring of physiological and biological parameters such as heart rate, blood pressure and blood glucose levels. This can be useful in various medical settings, where early detection and intervention can be critical. With the growth of technology trends and big data, these sensors are designed to be flexible, minimally invasive, comfortable to the skin and capable of wireless communication with other devices, including smartwatches, phones and tablets for displaying, post-analyzing and reporting to an encrypted server for the development of telemedicine protocols [11].

Epidermal wearable biosensors are likely to become even more advanced and sophisticated, allowing for more measurements to be adopted in healthcare settings and providing more reliable and accurate data. In 2022, the global market for biosensors was valued at USD 26.8 billion, and it is projected to grow significantly in the coming years at a compound annual growth rate (CAGR) of 8.0% from 2023 to 2030. The increasing ageing of the world’s population and the prevalence of chronic diseases, such as diabetes and cardiovascular diseases, is leading to a demand for wearable biosensors that can provide the continuous control of a person’s health status. Additionally, the rise in the popularity of fitness and wellness tracking devices is also contributing to the growth of this market.

Advanced research is mainly focusing on providing low cost, accessible and non-invasive devices, serving for a continuous and real-time sampling of the body fluids. One of the challenges of ISF extraction is the sensitivity of epidermal biosensors to detect biomarkers in low concentrations and small amounts of ISF. Furthermore, these devices aim to avoid skin damage by using collection techniques that are able to extract ISF without the irritation and contamination of the skin. There are also many challenges for ISF sampling and analysis using commercially available MNA devices, such as the biocompatibility of sensing components and the stability of sensing layer in vivo. Research to solve these challenges should focus on the preparation of MNA from biocompatible materials or the modification of the MNA surface.

Nowadays, wearable sensors could realize a large increase in both research and commercialization, due to their high performance for personalized health. In order to overcome the challenges, biosensors need to be integrated with machine learning, the internet of things, and communication for additional functionalities. Building flexible epidermal devices with high mechanical flexibility and electrochemical sensitivity requires more interventions, such as integrating a type of nonvolatile memories for information storage capability [198], microfluidic technology for preconcentration to solve low concentration challenges and nanotechnology for sensing mechanism advancements [11,199,200,201]. Fusing chemical, physical and electrophysiological sensors on the same platform can also help with the manufacture of hybrid wearable sensors, which offer a more comprehensive monitoring and understanding of the body’s state [199]. Some features can also be integrated, such as the implementation of new biorecognition elements and nucleic acids, as an example [202]. Finally, self-powered wearable sensors are a great future prospect leading toward personalized healthcare, covering biosensors, energy harvesters, energy storage and power supply strategies. Different methods can be manifested for generating power in wearable biosensors, which can be based on harvesting energy from human motion, body heat and ambient light. For this, several materials and technologies can be used to create self-powered biosensors, such as piezoelectric materials, thermoelectric materials and photovoltaic materials. Some examples of self-powered biosensors are a wearable device that uses body heat to power a heart rate monitor, a patch that harvests energy from the motion of the wearer’s body to power a glucose sensor and a bracelet that uses solar cells to generate power for a humidity sensor [203]. Moreover, the latest research has suggested a new wearable biosensor network that utilizes artificial intelligence (AI) to analyze the collected data. This biosensor network consists of multiple wearable devices that can collect data from various physiological signals. The data collected from these devices are then analyzed using machine learning algorithms to detect diseases and predict potential health issues [204].

## Figures and Tables

**Figure 1 micromachines-14-01452-f001:**
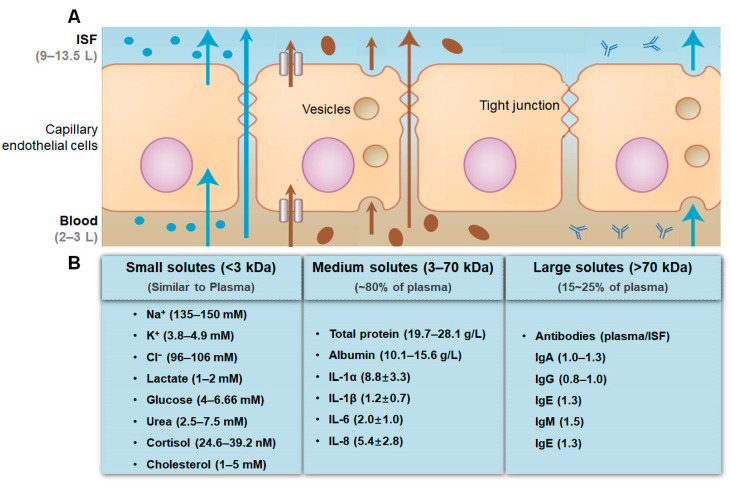
Schematic diagram of (**A**) three ways for biomarkers to enter ISF (adapted with permission from Ref. [8], copyright 2019, Springer Nature) and (**B**) the physiological concentrations of the common biomarkers in ISF [4,17,25,65,76,80,81,82].

**Figure 2 micromachines-14-01452-f002:**
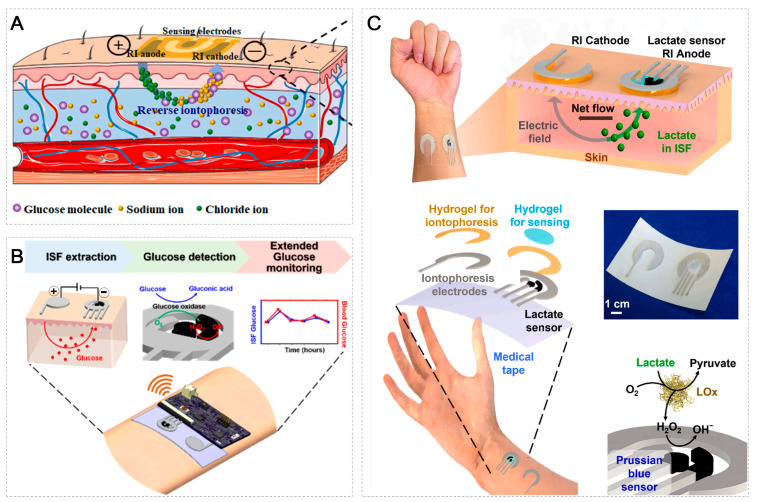
Reverse iontophoresis for ISF extraction. (**A**) Schematic showing a transdermal extraction of glucose based on RI with a screen-printed glucose biosensor (reprinted with permission from [115], copyright 2023, Elsevier). (**B**) A skin-worn electrochemical biosensor for the noninvasive monitoring of glucose in the ISF (reprinted with permission from [37], copyright 2021, American Chemical Society). (**C**) Concept of a non-invasive, wearable ISF lactate monitoring patch (reprinted with permission from [20], copyright 2023, Elsevier).

**Table 1 micromachines-14-01452-t001:** Epidermal wearable biosensor-based biofluids biomarkers analysis in clinical and preclinical applications.

Chronic Disease	Biomarkers	Biofluids					
		Characteristics	ISF	Sweat	Tear	Saliva	Serum
Cystic Fibrosis	Cl^−^	Con. ^1^ (mM)	96–106 [17,25]	10–90 [26]	120–135 [27]	6–35 [28]	96–106 [25]
		WP. ^2^	Epidermal patch [29]	Epidermal sticker [30]	Contact lens [31]	–	–
Diabetes	Glucose	Con. (mM)	4–6.66 [8]	0.02–0.6 [32,33]	0.05–0.5 [32,34]	0.03–0.08 [35,36]	4.44–6.66 [8]
		WP.	Epidermal patch [37]	Epidermal patch [38]	Contact lens [39]	Mouthguard sensor [40]	–
	Insulin	Con.	–	–	–	22–28 ^4^ 78–114 ^5^ pM [41]	2.6–31.1 μU/mL [42]
		WP.	Epidermal MNA patch [21]	–	–	–	–
Sepsis	Lactate	Con. (mM)	1–2 [17]	5–40 [5]	1–5 [27]	0.11 ± 0.02 [43]	0.5–1 [8]
		WP.	Epidermal MNA patch [44]	Bandage [45]	Wireless sensing system [46]	–	–
Gout	Uric acid	Con. (μM)	–	30–80 [47]	0.03–0.42 [48]	10–30 [49]	100–500 [50,51]
		WP.	Epidermal MNA patch [52]	Epidermal patch [53]	Contact lens [54]	Mouthguard sensor [55]	–
Breast cancer	ErbB2 ^3^	Con. (ng/mL)	–	–	–	0.5–44.7 [56]	2–15 [57]
		WP.	Epidermal MNA patch [58]	–	–	–	–
Preeclampsia	Estrogen	Con. (nM)	–	–	–	Positive [59]	31.5–44.6 [60]
		WP.	Epidermal MNA patch [61]	–	–	–	–
Neurodegeneration	H_2_O_2_	Con. (μM)	–	–	<200	–	1–5 [62]
		WP.	Epidermal MNA patch [63]	Epidermal patch [64]	–	–	–
Anxiety	Cortisol	Con. (nM)	24.6–39.2 [65]	0.66–7.73 [66]	2.76–110 [67]	7.7–14.0 [65]	2.76–8.28 [68]
		WP.	In vitro immunosensor [65]	Epidermal patch [69,70]	Contact lens [67]	–	–
Mood, Stress	Serotonin	Con. (nM)	–	–	3.4–21.5 [71]	7175–9804 [72]	30–170 [73,74]
		WP.	Epidermal MNA patch [21]	–	–	–	–

^1^ Con.: Concentration of healthy individuals. ^2^ WP.: Wearable Platform. ^3^ ErbB2: epidermal growth factor receptor 2. ^4^ Fasting salivary insulin level. ^5^ Swallowed meal.

**Table 3 micromachines-14-01452-t003:** Epidermal wearable biosensor-based ISF biomarker analysis in clinical and preclinical applications.

Bio- Markers	Related Disease	ISF Extraction Strategies	Related Materials	Sensing Techniques	Detection Range	Detection Limit	Application	Reference
Glucose	Diabetes	RI	Ag-G/CNTs ^1^ textile	Electrochemical	0–0.1 mM 1–30 mM	0.06 μM	Preclinical	[125]
		RI	PVA/BTCA/β-CD/GOx/AuNPs NF ^2^ hydrogels	Electrochemical	0–0.5 mM	0.01 mM	Preclinical	[191]
		MNA	Au-MWCNTs/pMB ^3^	Electrochemical	0.05–5 mM	7 μM	Preclinical	[44]
		MNA	Ag/AgCl	Electrochemical	2.5–22.5 mM	–	Preclinical	[86]
		MNA	Photopolymer	Colorimetric	0–10 mM	–	Preclinical	[192]
		MNA	Au/Pt-black/Nf	Electrochemical	1–30 mM	22 µM	Preclinical	[193]
		MD	AuNPs/Ag/AgCl	Electrochemical	0–9 mM	0.08 mM	Preclinical	[194]
		RI	Ag/AgCl	Electrochemical	0–22 mM	–	Preclinical	[37]
		MNA	PEGDA ^4^	Colorimetric	0–12 mM	–	Preclinical	[195]
Insulin	Diabetes	MNA	MeHA ^5^	Aptamer-based assay	0.1–3 nM	1.3 μM	Preclinical	[21]
Serotonin	Mood, sleep, digestion, wound healing, bone health, blood clotting	MNA	MeHA	Aptamer-based assay	0.5–4 μM	0.1 μM	Preclinical	[21]
Ketone bodies	Diabetic ketoacidosis	MNA	–	Electrochemical	1–10 mM	50 μM	Preclinical	[148]
		MNA	–	Electrochemical	0.1–2.4 mM	–	Preclinical	[196]
pH	Acute respiratory distress, peripheral artery disease, etc.	MNA	OrmoComp^®^ (Polymer)	Electrochemical	4.0–8.6	–	Preclinical	[18]
		MNA	PEGDA	Colorimetric	7.0–10.0	−	Preclinical	[195]
Lactate	Sepsis, malaria, dengue	MNA	Au-MWCNTs/pMB	Electrochemical	10–100 μM	3 μM	Preclinical	[44]
		MNA	Poly(carbonate)	Electrochemical	0–30 mM	−	Preclinical	[143]
		RI	Ag	Electrochemical	0–5 mM	0.15 mM	Preclinical	[20]
Uric acid	Gout	MNA	Poly(vinyl alcohol)	Colorimetric	200–1000 μM	65 μM	Preclinical	[52]
		MNA	Hyaluronic acid	Colorimetric	−	−	Preclinical	[85]
ErbB2	Breast cancer	MNA	Silicon	Electrochemical	10–250 ng/mL	4.8 ng/mL	Preclinical	[58]
Estrogen	Preeclampsia	MNA	Aluminum	Immunoassay	0.5–1000 ng mL^−1^	50 pg mL^−1^	Preclinical	[61]
Glycine	Multiple physiological functions	MNA	Stainless steel	Electrochemical	25–600 μM	7.9 μM	Preclinical	[197]
Levodopa	Parkinson management	MNA	MeHA	Electrochemical	10 nM–10 μM	100 nM	Preclinical	[22]
		MNA	Carbonpaste	Electrochemical	0.25–3 μM	0.25 μM	Preclinical	[88]
H_2_O_2_	Senescence, neurodegeneration, cancer	MNA	Steel	Electrochemical	0–6 mM	0.1 mM	Preclinical	[63]

^1^ Ag-G/CNTs, Ag deposited graphene (G) and carbon nanotubes. ^2^ PVA/BTCA/β-CD/GOx/AuNPs NF, poly(vinyl alcohol)/1,2,3,4-butanetetracarboxylic acid/β-cyclodextrin/glucose oxidase/gold nanoparticles nanofibers. ^3^ Au-MWCNTs/pMB, Au-multiwalled carbon nanotubes/polymethylene blue. ^4^ PEGDA, poly(ethylene glycol) diacrylate. ^5^ MeHA, methacrylated hyaluronic acid.

## Data Availability

Not applicable.

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
