# Peer review of "Epidermal Wearable Biosensors for the Continuous Monitoring of Biomarkers of Chronic Disease in Interstitial Fluid"

_micromachines, 2023, doi:10.3390/mi14071452_

Round 1

Reviewer 1 Report

In this review, the authors reviewed the various methods and applications of interstitial fluid sensing. As this is an important topic in the field of analytical chemistry and wearable sensors, many comprehensive reviews have been published on this very same topic. This review, although summarized many useful and correct information, in my opinion, did not differentiate itself with other recent reviews. Furthermore, I have the following comments that the authors may want to address to improve the quality of this work: 

1. The early introduction about POC devices is not related and unnecessary for this topic.

2. Ref 37 was a research paper on sweat sensing, which the authors cited as a paper on ISF sensing.

3. A figure that summarizes the common biomarkers and their corresponding physiological concentration in ISF would be helpful for readers. 

4. As a review paper to discuss the timely progress in this field, more than half of the references are published over 5 years ago. Some more mentions of the progress within the recent few years would improve the quality and novelty of this review. 

5. Many of the references to commercial progress in this field has been somewhat outdated too. For example, progress in algorithms from CGM companies such as Abbott and Dexcom was used to minimize the delay from ISF to blood; Many mentions of commercialized ISF monitoring devices did not refer to the functionalities of their newest generation;

6.  Although the authors defined the topic of this review to be mostly epidermal sensors, ISF sensing that uses invasive transdermal methods was also discussed. In this case, authors should also mention many of the subdermal implanted ISF sensors, some of which have already been commercialized.

Reviewer 2 Report

This review is interesting and worthy of publication in this journal, however, some concerns should be addressed before publication.

1.       In the introduction section, a comparison of other available wearable biosensors could be added.

2.       Although the authors already mentioned the benefit of the ISF method that provides more stability and reliability, more introduction to the advantages of the ISF method should be added.

3.       To avoid copyright issues, the sentences “reprinted with permission from [references], copyright [year] [publisher]” should be added to each figure that was reused from other studies.

4.       In section 3, although the authors explain each method in the subsection, additional figures such as Figure 3 for other methods could be helpful. 
